# The Stochastic Nature of Functional Responses

**DOI:** 10.3390/e23050575

**Published:** 2021-05-07

**Authors:** Gian Marco Palamara, José A. Capitán, David Alonso

**Affiliations:** 1Theoretical and Computational Ecology, Center for Advanced Studies of Blanes (CEAB-CSIC), Spanish Council for Scientific Research, Acces Cala St. Francesc 14, E-17300 Blanes, Spain; dalonso@ceab.csic.es; 2Complex Systems Group, Department of Applied Mathematics, Universidad Politécnica de Madrid, Av. Juan de Herrera 6, E-28040 Madrid, Spain; ja.capitan@upm.es

**Keywords:** stochastic consumer-resource dynamics, Holling type II and type III functional responses, Beddington–DeAngelis functional response, system’s size expansion, feeding experiments

## Abstract

Functional responses are non-linear functions commonly used to describe the variation in the rate of consumption of resources by a consumer. They have been widely used in both theoretical and empirical studies, but a comprehensive understanding of their parameters at different levels of description remains elusive. Here, by depicting consumers and resources as stochastic systems of interacting particles, we present a minimal set of reactions for consumer resource dynamics. We rigorously derived the corresponding system of ODEs, from which we obtained via asymptotic expansions classical 2D consumer-resource dynamics, characterized by different functional responses. We also derived functional responses by focusing on the subset of reactions describing only the feeding process. This involves fixing the total number of consumers and resources, which we call chemostatic conditions. By comparing these two ways of deriving functional responses, we showed that classical functional response parameters in effective 2D consumer-resource dynamics differ from the same parameters obtained by measuring (or deriving) functional responses for typical feeding experiments under chemostatic conditions, which points to potential errors in interpreting empirical data. We finally discuss possible generalizations of our models to systems with multiple consumers and more complex population structures, including spatial dynamics. Our stochastic approach builds on fundamental ecological processes and has natural connections to basic ecological theory.

## 1. Introduction

Models of predator-prey dynamics are the archetypal description of consumer-resource interaction. Since the pioneering work of Lotka and Volterra [1,2], coupled first-order ordinary differential equations to describe the temporal variation of the abundance of two species, a consumer and a resource, are used by ecologists in all sorts of applications [3,4,5,6,7,8,9]. In this context, the term functional response [10,11,12] was introduced to capture the variation in the rate of consumption of a resource by a consumer when the density of the resource changes. Functional responses measure how per capita average rates of resource consumption respond to the density of both resources and consumers. Such functions have been derived by isolating the feeding process, that is by considering that both resource and consumer densities remain constant. These responses can be summarized by plotting the number of resource items consumed per unit time by a consumer as a function of resource density. Holling [11,12] used the term type I to describe a linear relationship between feeding rates and resource density, while the term type II describes a non-linear relationship between feeding rates and resource density, where the slope of the curve monotonically decreases with increasing resource density, saturating at a constant value of resource consumption. Such a functional form is obtained assuming consumers move from a phase where they actively search for resources to a handling phase where they are occupied with consuming/digesting and not able to search for new resource items [13]. The term type III is associated with similar non-linear functions, where the slope first slightly increases, then, after a certain threshold, steeply increases, and finally, decreases, producing a sigmoidal shape in the curve [14]. It has been shown both theoretically [15] and empirically [16] that type III functional responses have a stabilizing effect on population dynamics. Assuming that consumers can also interfere with each other, e.g., by competing for the same resource item during searching and handling, the functional response becomes a function of both resource and consumer density. Non-linear functional forms in both consumer and resource density were independently proposed by Beddington [17] and DeAngelis [18]. The Beddington–DeAngelis functional response for predator interference has been further characterized and extended in subsequent studies [19,20,21,22].

Functional responses are typically measured in feeding experiments with a controlled resource density, which isolates the feeding process and enables quantifying resource intake. Several statistical methods to infer functional response parameters from feeding experiments are available [23,24,25]. Functional response parameters can also be directly inferred from consumer resource time series obtained from empirical studies of population dynamics [16,26,27,28]. Both empirical approaches and corresponding inference methods are based on the assumption that the parameterizations of functional responses obtained by feeding experiments are analogous to the parameterizations of the 2D ODEs describing consumer resource population dynamics. The type II functional response is still the most widely used functional response, and the corresponding differential equations for predator-prey dynamics, also known as the the Rosenzweig–MacArthur equations [29], have been widely studied in the dynamical systems literature [30,31,32]. Furthermore, the dynamical properties of consumer resource equations with the Beddington–DeAngelis functional response are well studied [33,34].

A complete review of the properties of these equations is beyond the scope of this paper. Instead, here, we underline how to derive such macroscopic equations from the elementary processes affecting the dynamics of individuals. Statistical physics has developed an array of powerful tools to scale up from microscopic to macroscopic dynamics [35,36]. These methods, originally developed to describe particle systems or chemical kinetics, can be more generally applied to many-body systems, and their use in population and community ecology has become widespread [37,38,39,40,41,42,43,44,45]. Examples of such derivations were given by Dawes and Souza [46], who proposed a minimal set of reactions to derive type I, II, and III functional responses, and by Van der Meer and Smallegange [21], who used similar arguments to derive a stochastic version of the Beddington–DeAngelis functional response. We provide here a simpler derivation that can be easily extended to include a more general set of reactions and accommodate different aspects of consumer resource interactions. We made the model general by allowing immigration and resource logistic growth and by defining the formation of a consumers-resource complex and triplets composed by two consumers fighting for one resource, which describes handling and interference, respectively. Although the stochastic description of the feeding process we provide here is not exhaustive, since we are not including direct consumer interference, it provides the basis for more elaborated reaction schemes. The paper is structured as follows

In Section 2, we give the description of a minimal set of reactions to define consumer-resource dynamics, including predators handling prey and predators interfering. We use the Van Kampen, systems size expansion to get, through the master equation of the discrete system, a mean field version of the model consisting in a set of four ODEs.In Section 3, we provide a detailed analysis of the dynamical properties of the 4D ODE system, including bifurcation analysis.In Section 4, we show how different arguments of time scales’ separation give rise to the classical 2D systems described in the ecological literature, i.e., predator-prey equations with Beddington–DeAngelis and Holling type II and type I functional responses.In Section 5, we derive classical functional responses (Holling type II and III and Beddington–DeAngelis) inspired by [14]. We emphasize their stochastic nature, by deriving the probability distribution of feeding events. By doing so, we clarify the assumptions that enable all these derivations. We obtained functional responses that differ in their parameterization and in their functional form (for the Beddington–DeAngelis case) from the functional responses obtained via separation of time scales in the previous section.In Section 6, we discuss potential limitations and possible generalizations of the model, describing future avenues of research.

## 2. A Minimal Set of Stochastic Processes

Here, we describe the dynamics of S=2 species, a resource $\phi_{R}$ and a consumer $X_{A}$. We characterize the dynamics distinguishing the two species by defining a maximum number of resource individuals N≫0 (N∈N) that can be packed in the local community. This parameter plays the role of the size of the system. This characterization introduces a limit in the total immigration rate of the resource species, $\Lambda_{R}$, between zero, when the system is full with *N* resource individuals, and *N*, when it is empty. This results in a density-dependent total immigration rate and a density-independent death rate.
(1)$$\begin{array}{ccc} \emptyset & \overset{\;\lambda_{R}\;}{\to } & \phi_{R}, \end{array}$$
(2)$$\begin{array}{ccc} \phi_{R} & \overset{\;\delta_{R}\;}{\to } & \emptyset , \end{array}$$
where $\delta_{R}$ is the intrinsic death rate of the resources and $\Lambda_{R}\left(t\right)=\lambda_{R}\left(N-n_{R}\left(t\right)\right)$. Note that we can always consider the system’s size as divided into *N* small portions or sites, which means that $\lambda_{R}$ can be interpreted as an immigration rate per empty site. Here, $n_{R}\left(t\right)$ is the number of resource individuals in the local community at time *t*. To further characterize resource dynamics, we considered another reaction describing birth events,
(3)$$\begin{array}{ccc} \phi_{R}+\emptyset & \overset{\;\beta \;}{\to } & \phi_{R}+\phi_{R}. \end{array}$$Here, the per capita resource rate is given by $B\left(t\right)=\beta \frac{N-n_{R}\left(t\right)}{N}$, and β can be regarded as a per capita birth rate when the system is almost empty.

The dynamics of the consumers is described by immigration and death reactions,
(4)$$\begin{array}{ccc} \emptyset & \overset{\;\lambda_{A}\;}{\to } & X_{A}, \end{array}$$
(5)$$\begin{array}{ccc} X_{A} & \overset{\;\delta_{A}\;}{\to } & \emptyset , \end{array}$$
and by processes associated with interactions between consumer and resource individuals describing, e.g., the feeding process with reactions: (6)$$\begin{array}{ccc} X_{A}+\phi_{R} & \overset{\;\alpha \;}{\to } & X_{\left[AR\right]}+\emptyset , \end{array}$$(7)$$\begin{array}{ccc} X_{\left[AR\right]}+\emptyset & \overset{\;\nu \;}{\to } & X_{A}+X_{A} \end{array}$$
where the formation of consumer-resource pairs $X_{\left[AR\right]}$ happens at a total rate $\frac{\alpha n_{R}n_{A}}{N}$ and the degradation of pairs into newly born consumers happens at rate $\nu n_{\left[AR\right]}$. In Reactions (6) and (7), α is the encounter rate between consumers and resources and ν is a degradation rate of pairs, defining the “handling time” $\tau_{H}:=1/\nu$ of individual consumers.

We can finally consider another process related to consumer interference formalized in triplet formation, given by reactions of the kind: (8)$$\begin{array}{c} X_{\left[AR\right]}+X_{A}\overset{\;\chi \;}{\to }X_{\left[ARA\right]}+\emptyset , \end{array}$$(9)$$\begin{array}{c} X_{\left[ARA\right]}+\emptyset \overset{\;\eta \;}{\to }X_{\left[AR\right]}+X_{A}, \end{array}$$
where χ is the rate of triplet formation that measures the propensity of free consumers to attack handling consumers and η is the rate of triplet degradation, which defines the “interference time” $\tau_{I}:=1/\eta$. Note that we did not consider interference between free consumers, i.e., reactions leading to the formation of consumer pairs $X_{\left[AA\right]}$.

### Derivation of Mean Field Approximations

The process defined by Reactions (1)–(9) can be formulated as a stochastic process. Fixing the number of sites to *N*, the system state is completely defined by a vector $n=(n_{R},n_{A},n_{\left[AR\right]},n_{\left[ARA\right]})$, and the dynamics is given by a discrete state Markov process, where $P(n,t|n_{0},t_{0})$ denotes the probability that the system is in state n at time t, given it was in state $n_{0}$ at time $t_{0}<t$.

The transition rates associated with elemental processes are: (10)$$\begin{array}{cc} \; & T_{R}^{+}:=T\left(n_{R}+1,n_{A},n_{\left[AR\right]},n_{\left[ARA\right]}|n\right)=\lambda_{R}\left(N-n_{R}\right)+\frac{\beta n_{R}\left(N-n_{R}\right)}{N-1}, \end{array}$$(11)$$\begin{array}{cc} \; & T_{R}^{-}:=T\left(n_{R}-1,n_{A},n_{\left[AR\right]},n_{\left[ARA\right]}|n\right)=\delta_{R}n_{R}, \end{array}$$(12)$$\begin{array}{cc} \; & T_{A}^{+}:=T\left(n_{R},n_{A}+1,n_{\left[AR\right]},n_{\left[ARA\right]}|n\right)=\lambda_{A}N, \end{array}$$(13)$$\begin{array}{cc} \; & T_{A}^{-}:=T\left(n_{R},n_{A}-1,n_{\left[AR\right]},n_{\left[ARA\right]}|n\right)=\delta_{A}n_{A}, \end{array}$$(14)$$\begin{array}{cc} \; & T_{\left[AR\right]}^{+}:=T\left(n_{R}-1,n_{A}-1,n_{\left[AR\right]}+1,n_{\left[ARA\right]}|n\right)=\frac{\alpha n_{A}n_{R}}{N}, \end{array}$$(15)$$\begin{array}{cc} \; & T_{\left[AR\right]}^{-}:=T\left(n_{R},n_{A}+2,n_{\left[AR\right]}-1,n_{\left[ARA\right]}|n\right)=\nu n_{\left[AR\right]}, \end{array}$$(16)$$\begin{array}{cc} \; & T_{\left[ARA\right]}^{+}:=T\left(n_{R},n_{A}-1,n_{\left[AR\right]}-1,n_{\left[ARA\right]}+1|n\right)=\frac{\chi n_{A}n_{\left[AR\right]}}{N}, \end{array}$$(17)$$\begin{array}{cc} \; & T_{\left[ARA\right]}^{-}:=T\left(n_{R},n_{A}+1,n_{\left[AR\right]}+1,n_{\left[ARA\right]}-1|n\right)=\eta n_{\left[ARA\right]}. \end{array}$$These transition rates are consistent with the set of reactions introduced above.

Let P(n,t) denote the conditional probability $P(n,t|n_{0},t_{0})$ obtained by imposing that the initial state, at $t=t_{0}$, is $n_{0}$, i.e., $P\left(n,t_{0}\right)=\delta \left(n-n_{0}\right)$. With this definition, the master equation reads as:(18)$$\begin{array}{cc} \frac{\partial P}{\partial t} & =\left(E_{R}^{-1}-I\right)\left(T_{R}^{+}P\right)+\left(E_{R}-I\right)\left(T_{R}^{-}P\right)\\ \; & +\left(E_{A}^{-1}-I\right)\left(T_{A}^{+}P\right)+\left(E_{A}-I\right)\left(T_{A}^{-}P\right)\\ \; & +\left(E_{R}E_{A}E_{\left[AR\right]}^{-1}-I\right)\left(T_{\left[AR\right]}^{+}P\right)+\left(E_{A}^{-2}E_{\left[AR\right]}-I\right)\left(T_{\left[AR\right]}^{-}P\right)\\ \; & +\left(E_{A}E_{\left[AR\right]}E_{\left[ARA\right]}^{-1}-I\right)\left(T_{\left[ARA\right]}^{+}P\right)+\left(E_{A}^{-1}E_{\left[AR\right]}^{-1}E_{\left[ARA\right]}-I\right)\left(T_{\left[ARA\right]}^{-}P\right), \end{array}$$
where we introduced the one-step operators:(19)$$\begin{array}{cc} \; & E_{X}f\left(\ldots ,n_{X},\ldots \right)=f\left(\ldots ,n_{X}+1,\ldots \right),\\ \; & E_{X}^{-1}f\left(\ldots ,n_{X},\ldots \right)=f\left(\ldots ,n_{X}-1,\ldots \right), \end{array}$$
to make the notation compact. Observe that operator *X* acts on population abundance $n_{X}$, with X∈{R,A,[AR],[ARA]}.

It can be shown, via a systematic Van Kampen expansion [35] of the master equation on the system’s size (*N*), that the stochastic model defined above yields a deterministic approximation as the leading term of the expansion, together with a Fokker–Planck equation describing the noise in the next-to-leading order [35]. The mean field version of the process associated with the general set of reactions (1)–(9) is given by a system of coupled ODEs given by: (20)$$\begin{array}{cc} \frac{dn_{R}}{dt} & =\lambda_{R}\left(N-n_{R}\right)-\delta_{R}n_{R}+\frac{\left(\beta N-\beta n_{R}-\alpha n_{A}\right)n_{R}}{N}, \end{array}$$(21)$$\begin{array}{cc} \frac{dn_{A}}{dt} & =\lambda_{A}N-\delta_{A}n_{A}+2\nu n_{\left[AR\right]}+\eta n_{\left[ARA\right]}-\frac{\left(\alpha n_{R}+\chi n_{\left[AR\right]}\right)n_{A}}{N}, \end{array}$$(22)$$\begin{array}{cc} \frac{dn_{\left[AR\right]}}{dt} & =\eta n_{\left[ARA\right]}-\nu n_{\left[AR\right]}+\frac{\left(\alpha n_{R}-\chi n_{\left[AR\right]}\right)n_{A}}{N}, \end{array}$$(23)$$\begin{array}{cc} \frac{dn_{\left[ARA\right]}}{dt} & =\frac{\chi n_{\left[AR\right]}n_{A}}{N}-\eta n_{\left[ARA\right]}. \end{array}$$

In Appendix A, we illustrate how the Van Kampen expansion for the master equation proceeds for the case in which no triplet formation is considered, i.e., when χ=0=η.

## 3. Stability Analysis

The ODE system defined by Equations (20)–(23) has a rich behavior in terms of its stability and the qualitative analysis of the equilibrium points it exhibits. We first analyzed the equilibrium points of the full ODE system and found a transcritical bifurcation when $\lambda_{A}=0$. The system has limit cycles that emerge after a Hopf bifurcation, which we show in a particular case in Section 3.1.

At equilibrium, (23) yields $\eta n_{\left[ARA\right]}=\frac{\chi n_{\left[AR\right]}n_{A}}{N}$. Substitution into the r.h.s. of (22) yields $\nu n_{\left[AR\right]}=\frac{\alpha n_{R}n_{A}}{N}$. Therefore, substitution of these two relations into the r.h.s. of (21) gives, at equilibrium,
(24)$$\lambda_{A}N-\delta_{A}n_{A}+\frac{\alpha n_{R}n_{A}}{N}=0,$$
which, together with (20), forms a non-linear system of degree three for equilibrium abundances $n_{A}$ and $n_{R}$. This system can be reduced to a cubic equation for $n_{R}$, which reads:(25)$$\begin{array}{cc} -f_{R}^{3}+\left(1-\lambda_{R}^{\prime }-\delta_{R}^{\prime }+\delta_{A}^{\prime }\right)f_{R}^{2}+ & \\ & +\left[\frac{\alpha }{\beta }\lambda_{A}^{\prime }+\left(\delta_{R}^{\prime }-1\right)\delta_{A}^{\prime }+\left(\delta_{A}^{\prime }+1\right)\lambda_{R}^{\prime }\right]f_{R}-\lambda_{R}^{\prime }\delta_{A}^{\prime }=0, \end{array}$$
where we made the definitions $\lambda_{A}^{\prime }:=\lambda_{A}/\alpha$, $\lambda_{R}^{\prime }:=\lambda_{R}/\beta$, $\delta_{A}^{\prime }:=\delta_{A}/\alpha$, and $\delta_{R}^{\prime }:=\delta_{R}/\beta$ and used the resource abundance scaled by the system’s size $f_{R}:=n_{R}/N$. However, for the sake of simplicity, in what follows, we focus on the case of the absence of the immigration of the predator, $\lambda_{A}=0$.

In this case, (24) yields a trivial solution, $n_{A}=0$, which implies trivially that $n_{\left[AR\right]}=0$ and $n_{\left[ARA\right]}=0$. Therefore, the r.h.s. of (20) yields a second-order equation,
(26)$$\lambda_{R}^{\prime }\left(1-f_{R}\right)-\delta_{R}^{\prime }f_{R}+\left(1-f_{R}\right)f_{R}=0.$$This equation can be solved for $f_{R}$,
(27)$$f_{R}^{\pm }=\frac{1}{2}\left(q-\lambda_{R}^{\prime }\pm \sqrt{{\left(\lambda_{R}^{\prime }-q\right)}^{2}+4\lambda_{R}^{\prime }}\right),$$
where we introduced the control parameter $q:=1-\delta_{R}^{\prime }=1-\frac{\delta_{R}}{\beta }$, which can be seen as the ratio between the intrinsic growth rate ($r=\beta -\delta_{R}$) and the birth rate (β) of resources. The solution $f_{R}^{-}$ is always negative, so we discarded it, and we obtained the first set of solutions $n_{R}=Nf_{R}^{+}$ and $n_{A}=n_{\left[AR\right]}=n_{\left[ARA\right]}=0$.

On the other hand, for $\lambda_{A}=0$, Equation (24) yields also the constant (*q*-independent) solution $n_{R}=\frac{N\delta_{A}}{\alpha }=N\delta_{A}^{\prime }$. Again, using the r.h.s. of (20), we obtained a non-trivial equilibrium solution for $n_{A}$,
(28)$$n_{A}=\frac{N\beta }{\alpha }\left(-\delta_{A}^{\prime }+\frac{\lambda_{R}^{\prime }}{\delta_{A}^{\prime }}+q-\lambda_{R}^{\prime }\right)$$
which, in turn, yields the equilibrium pair abundance,
(29)$$n_{\left[AR\right]}=\frac{N\beta }{\nu }\left(-{\delta_{A}^{\prime }}^{2}+\left(q-\lambda_{R}^{\prime }\right)\delta_{A}^{\prime }+\lambda_{R}^{\prime }\right),$$
and triplet abundance,
(30)$$n_{\left[ARA\right]}=\frac{N\beta^{2}\chi }{\eta \nu \delta_{A}^{\prime }}{\left({\delta_{A}^{\prime }}^{2}-\left(q-\lambda_{R}^{\prime }\right)\delta_{A}^{\prime }-\lambda_{R}^{\prime }\right)}^{2}.$$

Now, we focus on the two equilibrium abundances for the resource, $n_{R}^{\left(1\right)}=Nf_{R}^{+}$ and $n_{R}^{\left(2\right)}=N\delta_{A}^{\prime }$. These two solutions cross each other at:(31)$$q^{⋆}=\delta_{A}^{\prime }+\lambda_{R}^{\prime }-\frac{\lambda_{R}^{\prime }}{\delta_{A}^{\prime }}=\frac{\delta_{A}}{\alpha }+\frac{\lambda_{R}}{\beta }-\frac{\alpha \lambda_{R}}{\beta \delta_{A}},$$
which is basically a restriction between model parameters,
(32)$$1-\frac{\delta_{R}}{\beta }=\frac{\delta_{A}}{\alpha }+\frac{\lambda_{R}}{\beta }-\frac{\alpha \lambda_{R}}{\beta \delta_{A}}.$$ Then, using the Jacobian matrix of the system (20)–(23) evaluated at the two equilibria, it is easy to show that the solution for $n_{R}^{\left(2\right)}=N\delta_{A}^{\prime }$ is stable for $q<q^{⋆}$ and unstable for $q>q^{⋆}$. Consistently, the equilibrium solution for $n_{R}^{\left(1\right)}=Nf_{R}^{+}$ is unstable for $q<q^{⋆}$ and stable for $q>q^{⋆}$. Therefore, we found a transcritical bifurcation at $q=q^{⋆}$ for $\lambda_{A}=0$. Figure 1 shows the transcritical bifurcation as the stability of the steady-state changes by increasing the control parameter above the threshold $q^{⋆}$.

### 3.1. Hopf Bifurcation

We also observed the emergence of a Hopf bifurcation for several ranges of model parameters. Here, we illustrate the existence of a Hopf bifurcation for $\lambda_{R}=0$, $\lambda_{A}=0$, and $\delta_{R}=0$, in which we were able to calculate a series expansion of eigenvalues for small β. In order to derive analytical expressions in the simplest setting, we considered the model in the absence of triplets.

Let us focus on the equilibrium abundances $n_{R}=\frac{N\delta_{A}}{\alpha }$, $n_{A}=\frac{N\beta }{\alpha }\left(1-\frac{\delta_{A}}{\alpha }\right)$, and $n_{\left[AR\right]}=\frac{N\beta \delta_{A}}{\alpha \nu }\left(1-\frac{\delta_{A}}{\alpha }\right)$. The bifurcation arises around this equilibrium point. Observe that these densities are meaningful if $\alpha >\delta_{A}$. Notice that the threshold given by (32) reduces to $\alpha =\delta_{A}$ in the particular case we considered ($\lambda_{R}=0$, $\lambda_{A}=0$, and $\delta_{R}=0$). For $\alpha >\delta_{A}$, the solution $f_{R}^{\left(2\right)}=\delta_{A}/\alpha$ derived above is the stable one.

At that equilibrium point, the Jacobian matrix reads:(33)$$J=\left(\begin{array}{ccc} -\frac{\beta \delta_{A}}{\alpha } & -\delta_{A} & 0\\ -\beta \left(1-\frac{\delta_{A}}{\alpha }\right) & -2\delta_{A} & 2\nu \\ \beta \left(1-\frac{\delta_{A}}{\alpha }\right) & \delta_{A} & -\nu \end{array}\right).$$ For β=0, it is easy to see that the eigenvalues of *J* are λ=0 (double) and $\lambda =-2\delta_{A}-\nu$, so the equilibrium point is neutrally stable, and a limit cycle appears. We now expand these eigenvalues for β≪1. The characteristic polynomial is:(34)$$\lambda^{3}+\left(2\delta_{A}+\frac{\beta \delta_{A}}{\alpha }+\nu \right)\lambda^{2}+\left(\frac{3\delta_{A}+\nu }{\alpha }-1\right)\beta \delta_{A}\lambda +\left(\frac{\delta_{A}}{\alpha }-1\right)\beta \delta_{A}\nu =0.$$This cubic equation can be expanded in power series of β (for a similar expansion, we refer the reader to [47]). First, we obtained a series expansion for the double root λ=0, obtained when β=0. The perturbation analysis starts by setting $\lambda =a\beta^{b}$ and then trying to determine the factor *a* and exponent *b* at leading order. Substitution into (Equation 34) yields:(35)$$\begin{array}{cc} -a^{3}\beta^{3b}+a\beta^{1+b}\delta_{A}\left(1-\frac{3\delta_{A}+\nu }{\alpha }\right)+\beta \nu \delta_{A}\left(-1+\frac{\delta_{A}}{\alpha }\right) & \\ & -\frac{\delta_{A}}{\alpha }a^{2}\beta^{2b+1}-\left(2\delta_{A}+\nu \right)a^{2}\beta^{2b}. \end{array}$$Therefore, the smallest powers in β are obtained by choosing b=1/2. Canceling the leading terms (which turn out to be of the order of β), we can determine an expression for *a* as the solution of:(36)$$\delta_{A}\nu \left(-1+\frac{\delta_{A}}{\alpha }\right)-\left(2\delta_{A}+\nu \right)a^{2}=0,$$
i.e.,
(37)$$a=\pm \sqrt{\frac{\delta_{A}\nu \left(\delta_{A}-\alpha \right)}{\alpha (2\delta_{A}+\nu )}}.$$
Here, we see that for $\alpha >\delta_{A}$, the double eigenvalue λ=0 obtained for β=0 splits into two complex eigenvalues with the imaginary part equal to:(38)Im(λ)=±βδAν(α−δA)α(2δA+ν).We can calculate the real part (at the lowest order in the β series expansion) by setting:(39)$$\lambda =\pm \sqrt{\frac{\beta \delta_{A}\nu \left(\alpha -\delta_{A}\right)}{\alpha (2\delta_{A}+\nu )}}i+c\beta .$$Expanding the characteristic polynomial up to order $\beta^{3/2}$ and setting equal to zero the coefficient of the leading term yields the following expression for *c*:(40)$$c=-\frac{\delta_{A}\left(\nu^{2}+2\left(3\delta_{A}-\alpha \right)\left(\delta_{A}+\nu \right)\right)}{2\alpha {\left(2\delta_{A}+\nu \right)}^{2}}.$$Therefore, for β≪1, we obtained the two complex conjugate eigenvalues:(41)$$\lambda =-\frac{\beta \delta_{A}\left(\nu^{2}+2\left(3\delta_{A}-\alpha \right)\left(\delta_{A}+\nu \right)\right)}{2\alpha {\left(2\delta_{A}+\nu \right)}^{2}}\pm \sqrt{\frac{\beta \delta_{A}\nu \left(\alpha -\delta_{A}\right)}{\alpha (2\delta_{A}+\nu )}}i+O\left(\beta^{3/2}\right).$$By setting $\nu^{2}+2\left(3\delta_{A}-\alpha \right)\left(\delta_{A}+\nu \right)=0$, we found the threshold of a Hopf bifurcation, which can be expressed as:(42)$$\alpha_{c}=3\delta_{A}+\frac{\nu^{2}}{2(\delta_{A}+\nu )}.$$Observe that $\alpha_{c}>\delta_{A}$, so eigenvalues stemming from the splitting of λ=0 (for β=0) are complex. For $\alpha >\alpha_{c}$, Re(λ)>0, and for $\delta_{A}<\alpha <\alpha_{c}$, Re(λ)<0 in the limit β≪1. Therefore, the eigenvalues cross over the imaginary axis, and a Hopf bifurcation arises: for $\alpha <\alpha_{c}$, eigenvalues have a negative real part, which leads to stable oscillations around the equilibrium point. Once the threshold $\alpha_{c}$ is crossed over ($\alpha >\alpha_{c}$), damped oscillations (with Re(λ)<0) no longer exist, and the equilibrium point is unstable. The dynamics converges to a limit cycle (i.e., we found a supercritical Hopf bifurcation as α increases).

For the sake of completeness, we provide here the series expansion for the third eigenvalue, which can be obtained in a similar way:(43)$$\lambda =-2\delta_{A}-\nu -\frac{2\beta \delta_{A}\left(\alpha -\delta_{A}\right)\left(\delta_{A}+\nu \right))}{\alpha {\left(2\delta_{A}+\nu \right)}^{2}}+O\left(\beta^{3/2}\right).$$Notice that this eigenvalue remains negative for small perturbations in β>0.

Figure 2 shows the stability of equilibrium solutions for arbitrary values of α and β. The diagram shows the transcritical bifurcation threshold (Equation 32), which separates the stable solution (1), where only the resource survives, from the coexistence equilibrium (2). This solution is first asymptotically stable, and then, damped oscillations around this stable equilibrium point arise. The rightmost threshold line in Figure 2 stands for the Hopf bifurcation, which separates stable oscillations from limit cycles.

## 4. Derivation of Functional Responses via the Separation of Time Scales

In this section, we discuss the asymptotic dynamics of the mean field description of the system (20)–(23). To simplify the calculations, we assumed there is no immigration in the system ($\lambda_{R}=0$ and $\lambda_{A}=0$). A simple asymptotic derivation can be made for the type I and type II functional responses. In this case, there is no interference between predators (χ=0, η=0), and we can write the mean field Equations (20)–(23) as: (44)$$\begin{array}{cc} \frac{dn_{R}}{dt} & =rn_{R}\left(1-\frac{n_{R}}{K}\right)-\frac{\alpha n_{R}n_{A}}{N}, \end{array}$$(45)$$\begin{array}{cc} \frac{dn_{A}}{dt} & =2\nu n_{\left[AR\right]}-\delta_{A}n_{A}-\frac{\alpha n_{R}n_{A}}{N}, \end{array}$$(46)$$\begin{array}{cc} \frac{dn_{\left[AR\right]}}{dt} & =\frac{\alpha n_{R}n_{A}}{N}-\nu n_{\left[AR\right]}, \end{array}$$
where $r=\beta -\delta_{R}$ and K=rN/β are the growth rate and carrying capacity of the resource. Assuming that compounds disappear very fast from the system, the time scale of consumers $X_{A}$ is longer than the time scale of the compounds $X_{\left[AR\right]}$. In this regime, putting $\frac{dn_{\left[AR\right]}}{dt}=0$ in Equation (46), we obtained $\nu n_{\left[AR\right]}=\alpha n_{A}n_{R}/N$, which we can substitute in Equation (45) to get: (47)$$\begin{array}{cc} \frac{dn_{R}}{dt} & =rn_{R}\left(1-\frac{n_{R}}{K}\right)-\frac{\alpha n_{R}n_{P}}{N}, \end{array}$$(48)$$\begin{array}{cc} \frac{dn_{P}}{dt} & =\frac{\alpha n_{R}n_{P}}{N}-\delta_{A}n_{P}, \end{array}$$
which are the classical predator-prey Lotka–Volterra equations [1,2], i.e., consumer-resource equations with a type I functional response. Note that in Equation (48), the numerical response, i.e., the per capita rate of food intake of the predator [48], is equivalent to the functional response. In other words, the conversion efficiency of resource density into predator density is one. In this case, the parameter α can be interpreted as the macroscopic attack rate of the predator, whose density is given by $n_{P}=n_{A}$.

In the opposite limit, when the time scale of compounds $X_{\left[AR\right]}$ is longer than the time scale of consumers $X_{A}$, we can assume that the degradation rate of compounds and the growth rate of resources are small compared to all the other parameters. Following [46], these assumptions can be summarized by setting $\nu =\epsilon \tilde{\nu }$ and $r=\epsilon \tilde{r}$ with 0<ϵ≪1, while $\tilde{\nu }$ and $\tilde{r}$ remain O(1). For any positive parameters, Equations (44)–(46) have simpler dynamics in the leading order. Making a simple rescaling in the time of the dynamics by writing $d/dt=\epsilon d/d\tilde{t}$ and substituting into Equation (21), we can now derive an expression for $n_{A}$ as:(49)$$n_{A}=\frac{2\epsilon \tilde{\nu }n_{\left[AR\right]}}{\delta_{A}+\alpha n_{R}/N}+O\left(\epsilon^{2}\right),$$
which can be substituted into Equations (20) and (22) to obtain an ODE system of two equations given by: (50)$$\begin{array}{cc} \frac{dn_{R}}{d\tilde{t}} & =\tilde{r}n_{R}\left(1-\frac{n_{R}}{K}\right)-\frac{\bar{\alpha }n_{R}n_{P}}{1+\bar{\alpha }{\bar{\tau }}_{H}n_{R}}+O\left(\epsilon \right), \end{array}$$(51)$$\begin{array}{cc} \frac{dn_{P}}{d\tilde{t}} & =\frac{\bar{\alpha }n_{R}n_{P}}{1+\bar{\alpha }{\bar{\tau }}_{H}n_{R}}-\delta_{P}n_{P}+O\left(\epsilon \right), \end{array}$$
which is the classical Rosenzweig–MacArthur model for predator-prey dynamics [29], where the total density of predators is slaved to the total number of compounds $n_{P}:=n_{A}+n_{\left[AR\right]}=n_{\left[AR\right]}+O\left(\epsilon \right)$. Equations (50) and (51) describe predator-prey dynamics characterized by logistic growth for the prey and a typical type II functional response with attack rate $\bar{\alpha }=2\alpha \tilde{\nu }/\delta_{A}$, handling time ${\bar{\tau }}_{H}=1/2\tilde{\nu }$, and death rate $\delta_{P}=\tilde{\nu }$. In this case as well, the numerical response in Equation (51) is given by the functional response of Equation (48). Note also that in this case, the parameters of the macroscopic equations can be expressed as a function of the parameters of the microscopic process (2)–(7), but are not intuitively related to the same parameters obtained considering typical heuristic arguments to derive functional responses [25].

Similar slaving arguments can be made for the general system (20)–(23), where also the degradation rate of triplets is small compared to the other rates (i.e, we additionally set $\eta =\epsilon^{2}\tilde{\eta }$). The derivation goes as follows.

Eliminating $n_{A}$ from Equation (21), we obtained, up to first order in ϵ,
(52)$$n_{A}=\frac{2\epsilon \tilde{\nu }n_{\left[AR\right]}}{\delta_{A}+\alpha n_{R}/N+\chi n_{\left[AR\right]}/N}+O\left(\epsilon^{2}\right),$$
which can be substituted into (20) to get an equation for the resource,
(53)$$\frac{dn_{R}}{d\tilde{t}}=\tilde{r}n_{R}\left(1-\frac{n_{R}}{K}\right)-\frac{2\alpha \tilde{\nu }n_{R}n_{\left[AR\right]}}{N\delta_{A}+\alpha n_{R}+\chi n_{\left[AR\right]}}+O\left(\epsilon \right).$$On the other hand, summing Equations (22) and (23) yields:(54)$$\frac{d}{d\tilde{t}}\left(n_{\left[AR\right]}+n_{\left[ARA\right]}\right)=\frac{2\alpha \tilde{\nu }n_{R}n_{\left[AR\right]}}{N\delta_{A}+\alpha n_{R}+\chi n_{\left[AR\right]}}-\tilde{\nu }n_{\left[AR\right]}+O\left(\epsilon \right).$$If we assumed that $n_{\left[ARA\right]}$ is proportional to $n_{\left[AR\right]}$ along the dynamics, say $n_{\left[ARA\right]}=\kappa n_{\left[AR\right]}$, and we defined the total density of consumers as $n_{P}:=n_{\left[AR\right]}+n_{\left[ARA\right]}=\left(1+\kappa \right)n_{\left[AR\right]}$, we finally obtained the typical functional form of the Beddington–DeAngelis functional response [19], including predator handling and interference,
(55)$$\begin{array}{cc} \frac{dn_{R}}{d\tilde{t}} & =\tilde{r}n_{R}\left(1-\frac{n_{R}}{K}\right)-\frac{\bar{\alpha }n_{R}n_{P}}{N\delta_{A}+\alpha n_{R}+\bar{\chi }n_{P}}+O\left(\epsilon \right), \end{array}$$
(56)$$\begin{array}{cc} \frac{dn_{P}}{d\tilde{t}} & =\frac{\bar{\alpha }n_{R}n_{P}}{N\delta_{A}+\alpha n_{R}+\bar{\chi }n_{P}}-{\bar{\delta }}_{P}n_{P}+O\left(\epsilon \right), \end{array}$$
where the effective interference rate is given by $\bar{\chi }=\frac{\chi }{1+\kappa }$, the effective attack rate is $\bar{\alpha }=\frac{2\alpha \tilde{\nu }}{1+\kappa }$, and the effective death rate of predators is given by ${\bar{\delta }}_{P}=\frac{\tilde{\nu }}{1+\kappa }$. However, we did not find a straightforward way to determine κ in terms of the parameters of the original dynamics given by Equations (20)–(23). In addition, we had to assume that densities $n_{\left[AR\right]}$ and $n_{\left[ARA\right]}$ are proportional along the temporal dynamics to obtain (55)–(56).

## 5. Stochastic Feeding Rates

A rigorous definition of a predator functional response can be precisely given as the number of resource units an average individual predator is able to consume per unit time. Since predator consumption rates respond in principle to resource density, any empirical approach intending to measure these rates will require fixing both resource and predator densities. We call these conditions chemostatic, in analogy with experiments done in chemostats, i.e., reactors where nutrient concentrations and density of microorganisms can be controlled [49]. Given the processes that define the stochastic dynamics in Section 2, we can now ask what type of functional response emerges under chemostatic conditions.

Therefore, let us assume that we maintain both the number of resource units $n_{R}^{0}$ and the total number of consumers $n_{A}^{0}=n_{A}+n_{\left[AR\right]}$ constant. Let us also assume first no formation of triplets and focus only on the feeding process. In this case, the dynamics can be simply described by the following two processes: (57)$$\begin{array}{ccc} X_{A}+\phi_{R} & \overset{\;\alpha \;}{\to } & X_{\left[AR\right]}+\emptyset , \end{array}$$(58)$$\begin{array}{c} X_{\left[AR\right]}\overset{\;\nu \;}{\to }X_{A}, \end{array}$$
where the first equation is tightly coupled to the addition of a new resource unit in order to keep resources fixed to exactly the same level $n_{R}^{0}$ all the time.

In order to calculate the functional response, i.e., the number of resource units captured by an individual predator per unit time, we introduced a new stochastic variable $R_{T}$, as the number of resource units consumed by a total population of consumers $n_{A}^{0}$, in a time interval, *T*. Then, the functional response, as a per capita feeding rate, over a certain period *T* and a total number of consumers $n_{A}^{0}$, becomes:(59)$$f\left(n_{R}^{0},n_{A}^{0};T\right)=\frac{1}{T}\frac{R_{T}}{n_{A}^{0}}.$$

This functional response is a stochastic per capita feeding rate. In principle, it is assumed to be both prey and predator dependent. It is described by the probability distribution of cumulative feeding events *n*, realized by the total number of consumers $n_{A}^{0}$, between Time 0 and *T*. If we characterize the configuration of the system by *n*, the number of accumulated feeding events, and $n_{A}$, the number of free predators ready to attack resources at any given time, the stochastic dynamics of the feeding process is governed by the following total transition rates: (60)$$\begin{array}{cc} \; & r_{n,n_{A}}:=T\left[\left(n+1,n_{A}-1\right)|\left(n,n_{A}\right)\right]=\alpha \frac{n_{R}^{0}}{N}\;n_{A}, \end{array}$$(61)$$\begin{array}{cc} \; & g_{n,n_{A}}:=T\left[\left(n,n_{A}+1\right)|\left(n,n_{A}\right)\right]=\nu \left(n_{A}^{0}-n_{A}\right). \end{array}$$

Equations (60) and (61) represent the rate at which the individual-based reactions (57)–(58) occur. Compare these to reactions (6)–(7). Notice that here we did not consider consumer growth nor resource depletion. When a consumer individual attacks a resource item, this item is instantaneously replenished in order to maintain the resource density constant. These rates can be then used to write a master equation (see Equation (A14) in Appendix B), which governs the temporal evolution for the probability of having an accumulated total of *n* resource items being consumed and $n_{A}$ free consumers at time *t*, $P(n,n_{A};t)$. This one-step process is linear in $n_{A}$ since both $n_{A}^{0}$ and $n_{R}^{0}$ are kept constant. Therefore, the probability distribution $P(n,n_{A};t)$, can be calculated exactly. The probability distribution characterizing the stochastic variable $R_{T}$ is, in fact, its marginal, i.e., the probability of having a given number *n* of accumulated feeding events at time *T*, regardless how many free consumers $n_{A}$ are around. It will be given by:(62)$$P\left(n,t\right)=\sum_{n_{A}=0}^{n_{A}^{0}}P\left(n,n_{A},t\right).$$

Before giving more details about the calculation of both the full probability distribution $P(n,n_{A};t)$ and its marginal, P(n,t), we first analyzed the average feeding rate, $\langle f\left(n_{R}^{0},n_{A}^{0};T\right)\rangle$. It is clear that the average total number of resource units $R\equiv \langle R_{T}\rangle$, consumed by a population of consumers monotonically increases in time according to:(63)$$\frac{dR}{dt}=\alpha \frac{n_{R}^{0}}{N}n_{A},$$
which, by defining:(64)$$\theta :=\alpha \frac{n_{R}^{0}}{N},$$
can also be written in integral form as:(65)$$R\left(T\right)=\theta \int_{0}^{T}n_{A}\left(t\right)dt.$$Therefore, the average feeding rate per individual consumer over a period *T* can be written as:(66)$$\langle f\left(n_{R}^{0},n_{A}^{0};T\right)\rangle =\frac{1}{T}\frac{\theta \int_{0}^{T}n_{A}\left(t\right)dt}{n_{A}^{0}}.$$

Since the total number of consumers is kept constant ($n_{A}^{0}=n_{A}+n_{\left[AR\right]}$), the average deterministic rate equations describing the feeding processes as defined in Reactions (57) and (58) can be reduced to a single equation for $n_{A}$,
(67)$$\frac{dn_{A}}{dt}=\nu n_{A}^{0}-\left(\theta +\nu \right)n_{A}.$$If we assume as initial condition $n_{A}\left(0\right)=n_{A}^{0}$, i.e., all consumers are free and ready to feed at time t=0, then we can integrate the last equation from zero to *T* as:(68)$$n_{A}\left(t\right)=\frac{\nu }{\nu +\theta }\left[1+\frac{\theta }{\nu }e^{-(\theta +\nu )t}\right]n_{A}^{0},$$
and therefore:(69)$$\int_{0}^{T}n_{A}\left(t\right)dt=\frac{\nu }{\nu +\theta }\left[T+\frac{\theta }{\nu (\nu +\theta )}\left(1-e^{-(\theta +\nu )T}\right)\right]n_{A}^{0}.$$Now, going back to Equation (Equation 66), we can write:(70)$$\langle f\left(n_{R}^{0},n_{A}^{0};T\right)\rangle =\frac{\theta \nu }{\nu +\theta }+\frac{\theta }{T\nu (\nu +\theta )}\left(1-e^{-(\theta +\nu )T}\right).$$This average rate per individual consumer tends to a stationary value as *T* tends to infinity, which is:(71)$$\langle f\left(n_{R}^{0}\right)\rangle =\frac{\theta \nu }{\nu +\theta }\;,$$
where we removed the time (to denote the asymptotic limit) and the consumer dependence. After introducing again θ, as defined in Equation (78), it is clear that the average consumption rate per individual consumer corresponds exactly to the typical Holling type II functional response, where ν is the inverse of the handling time and α is the attack rate:(72)$$\langle f\left(n_{R}^{0}\right)\rangle =\frac{\alpha \frac{n_{R}^{0}}{N}}{1+\frac{\alpha }{\nu }\frac{n_{R}^{0}}{N}}.$$

See also the red curve in Figure 3. In an empirical setting, where we monitored the total number of resource units consumed by a controlled population of $n_{A}^{0}$ consumers under chemostatic conditions over a period *T*, we would obtain a different total number for each experimental replicate. The scheme given by the reactions (57) and (58) predicts a theoretical distribution that can then be compared to data. We give details about the calculation of this distribution in Appendix B. In short, the transition rates given by Equations (60) and (61) are first used to write a master equation from which an equation for the probability generating function can be derived:(73)$$\frac{\partial G(x,y;t)}{\partial t}=\nu n_{A}^{0}\left(y-1\right)G\left(x,y;t\right)+\left[\theta (x-y)-\nu y(y-1)\right]\frac{\partial G(x,y;t)}{\partial y},$$
where the precise definition of G(x,y,t) is given in Equation (58) in Appendix B. This PDE is then solved under the initial condition $G\left(x,y,0\right)=y^{n_{A}^{0}}$ by the method of the characteristics, with the normalization condition G(0,0,t)=1, which yields the functional form:(74)$$G\left(x,y;t\right)=e^{\nu n_{A}^{0}\left(y-1\right)t}{\left(\frac{y_{0}^{+}\left(x\right)e^{-\Delta (x)t}-y_{0}^{-}\left(x\right)\frac{y-y_{0}^{+}\left(x\right)}{y-y_{0}^{-}\left(x\right)}}{e^{-\Delta (x)t}-\frac{y-y_{0}^{+}\left(x\right)}{y-y_{0}^{-}\left(x\right)}}\right)}^{n_{A}^{0}},$$
where $y_{0}^{-}\left(x\right)$, $y_{0}^{+}\left(x\right)$, and Δ(x) are functions of *x* (see Appendix B). It can be checked that this expression satisfies G(0,0,t)=1, for any *t*, and $G\left(x,y;0\right)=y^{n_{A}^{0}}$ as required. In addition, this allows the immediate calculation of the probability generating function of the marginal distribution P(n;t) by simply evaluating the full probability generating function at y=1:(75)$$G\left(x,1;t\right)={\left(\frac{y_{0}^{+}\left(x\right)\;e^{-\Delta (x)t}-y_{0}^{-}\left(x\right)\;\frac{1-y_{0}^{+}\left(x\right)}{1-y_{0}^{-}\left(x\right)}}{e^{-\Delta (x)t}-\frac{1-y_{0}^{+}\left(x\right)}{1-y_{0}^{-}\left(x\right)}}\right)}^{n_{A}^{0}}.$$

The distribution P(n;t) gives the probability of *n* prey items disappearing under the pressure of a constant total number of consumers $n_{A}^{0}$, from Time 0 until time *t*. Under controlled experimental conditions [24], this probability can be used as the exact likelihood function for inference purposes. See also [50] and the references therein, for a broader description of the challenges of the experiments and the inference of functional response parameters. Figure 3 shows stochastic simulations of the per capita rate compared to the Holling type II average provided by (Equation 72).

Note that the Holling type II functional response in Equation (Equation 72) naturally arises as a per capita average asymptotic feeding rate after reaching stationarity under chemostatic conditions. If the feeding mechanism differs from the simple one assumed by the reactions (57) and (58), other functional responses will be obtained. For instance, if consumer feeding rates are strongly affected by the density of the resources, we can describe feeding dynamics by the following two processes: (76)$$\begin{array}{ccc} X_{A}+n\phi_{R} & \overset{\;\alpha \;}{\to } & X_{\left[AR\right]}+\left(n-1\right)\phi_{R}, \end{array}$$(77)$$\begin{array}{c} X_{\left[AR\right]}\overset{\;\nu \;}{\to }X_{A}. \end{array}$$If we repeat an analogous derivation as for Holling type II, we arrived formally at the same Equation (Equation 67), but instead, parameter θ should now be defined as:(78)$$\theta :=\alpha {\left(\frac{n_{R}^{0}}{N}\right)}^{n}$$
which leads to a Holling type III functional response:(79)$$\langle f\left(n_{R}^{0}\right)\rangle =\frac{\alpha {\left(\frac{n_{R}^{0}}{N}\right)}^{n}}{1+\frac{\alpha }{\nu }{\left(\frac{n_{R}^{0}}{N}\right)}^{n}}.$$Note that Reaction (Equation 76) can be regarded as if the presence of n−1 extra resources facilitated the attack by consumers. Resources would have a self-catalytic effect on their own loss. In nature, it is plausible that when resources are clumped together, consumers encounter them better and faster. However, a Holling type III of exact order *n* (Equation (Equation 79) represents here of course an idealization.

Furthermore, let us now assume consumer interference through the formation of triplets. Then, feeding dynamics is specified by the following scheme: (80)$$\begin{array}{ccc} X_{A}+\phi_{R} & \overset{\;\alpha \;}{\to } & X_{\left[AR\right]}+\emptyset , \end{array}$$(81)$$\begin{array}{c} X_{\left[AR\right]}\overset{\;\nu \;}{\to }X_{A}. \end{array}$$(82)$$\begin{array}{c} X_{\left[AR\right]}+X_{A}\overset{\;\chi \;}{\to }X_{\left[ARA\right]}+\emptyset , \end{array}$$(83)$$\begin{array}{c} X_{\left[ARA\right]}+\emptyset \overset{\;\eta \;}{\to }X_{\left[AR\right]}+X_{A}, \end{array}$$Again, under chemostatic conditions, the resource level $n_{R}^{0}$ and the total number of consumers $n_{A}^{0}$ are both kept constant. In order to calculate the functional response as a per capita feeding rate over certain period *T*, $\langle f\left(n_{R}^{0},n_{A}^{0};T\right)\rangle$, we made use again of the definition given by Equation (Equation 66). The total consumer population is kept constant ($n_{A}^{0}=n_{A}+n_{\left[AR\right]}+2n_{\left[ARA\right]}$) and is distributed among the three types: free consumers $n_{A}$, handling/feeding consumers $n_{\left[AR\right]}$, and interfering consumers engaged in triplet formation $n_{\left[ARA\right]}$. Once this distribution reaches the steady state, there will be a steady number of consumers $n_{A}^{⋆}$, free to attack and deplete resources. Therefore, at stationarity, Equation (Equation 66) becomes:(84)$$\langle f\left(n_{R}^{0},n_{A}^{0}\right)\rangle =\theta \;\frac{n_{A}^{⋆}}{n_{A}^{0}},$$
where we dropped the dependence of the averaging time interval *T*, because this is an asymptotic rate and θ, defined again as in Equation (Equation 64), is a constant parameter, since $n_{R}^{0}$ is assumed constant. The value of $n_{A}^{⋆}$ is defined by the steady state that emerges from the rate equations describing the feeding mechanism depicted in the reaction scheme (80)–(83). One can check that this system should read as: (85)$$\begin{array}{cc} \frac{dn_{A}}{dt} & =\nu n_{\left[AR\right]}+\eta n_{\left[ARA\right]}-\alpha \frac{n_{R}^{0}}{N}\;n_{A}-\chi \frac{n_{\left[AR\right]}}{N}\;n_{A}, \end{array}$$(86)$$\begin{array}{cc} \frac{dn_{\left[AR\right]}}{dt} & =\eta n_{\left[ARA\right]}-\nu n_{\left[AR\right]}+\alpha \frac{n_{R}^{0}}{N}\;n_{A}-\chi \frac{n_{\left[AR\right]}}{N}\;n_{A}, \end{array}$$(87)$$\begin{array}{cc} \frac{dn_{\left[ARA\right]}}{dt} & =\chi \frac{n_{\left[AR\right]}}{N}\;n_{A}-\eta n_{\left[ARA\right]}. \end{array}$$In order to determine the steady state of these feeding dynamics, we better work with densities $f_{A}=n_{A}/N$, $f_{\left[AR\right]}=n_{\left[AR\right]}/N$, and $f_{\left[ARA\right]}=n_{\left[ARA\right]}/N$ and make use of the constraint of a constant consumer population ($n_{A}^{0}=n_{A}+n_{\left[AR\right]}+2n_{\left[ARA\right]}$), which reduces the ODE system to the following two equations: (88)$$\begin{array}{cc} \frac{df_{A}}{dt} & =\nu f_{\left[AR\right]}+\frac{\eta }{2}\left(f_{A}^{0}-f_{A}-f_{\left[AR\right]}\right)-\theta \;f_{A}-\chi f_{\left[AR\right]}\;f_{A}, \end{array}$$(89)$$\begin{array}{cc} \frac{df_{\left[AR\right]}}{dt} & =\frac{\eta }{2}\left(f_{A}^{0}-f_{A}-f_{\left[AR\right]}\right)-\nu f_{\left[AR\right]}+\theta \;f_{A}-\chi f_{\left[AR\right]}\;f_{A}. \end{array}$$By making these two equations equal to zero, we obtained steady state densities. In particular, the density of free consumers $f_{A}^{⋆}$ can be written as:(90)$$f_{A}^{⋆}=\frac{1}{4}\left[-\frac{\eta }{\chi }\left(1+\frac{\nu }{\theta }\right)+\sqrt{{\left(\frac{\eta }{\chi }\left(1+\frac{\nu }{\theta }\right)\right)}^{2}+8\frac{\eta }{\chi }\frac{\nu }{\theta }f_{A}^{0}}\right].$$Using this expression in Equation (Equation 84) and rearranging, we finally obtained the following per capita feeding rate of consumers at steady state given by:(91)$$\langle f\left(n_{R}^{0},n_{A}^{0}\right)\rangle =\frac{2\alpha \;\frac{n_{R}^{0}}{N}}{1+\frac{\alpha }{\nu }\frac{n_{R}^{0}}{N}+\sqrt{{\left(1+\frac{\alpha }{\nu }\frac{n_{R}^{0}}{N}\right)}^{2}+8\frac{\chi }{\eta }\frac{\alpha }{\nu }\frac{n_{R}^{0}}{N}\frac{n_{A}^{0}}{N}}}.$$Note that this is a predator-dependent functional response that generalizes Holling type II. If no triplet formation is considered (χ=0), we recovered Equation (Equation 72). Here, we considered consumers interfering with other consumers that have already caught a resource item, perhaps in the hope of getting a share. However, other possible forms of predator interference are possible. Figure 4 shows the comparison with stochastic simulations.

The functional response given by (Equation 91) reduces almost to a Beddington–DeAngelis form in the limit $\frac{\chi }{\eta }\ll 1$ and $\frac{\alpha }{\nu }\ll 1$. In this case, (Equation 91) reduces to:(92)$$\langle f\left(n_{R}^{0},n_{A}^{0}\right)\rangle \approx \frac{\alpha \;\frac{n_{R}^{0}}{N}}{1+\frac{2\alpha }{\nu }\frac{n_{R}^{0}}{N}+\frac{2\chi \alpha }{\eta \nu }\frac{n_{R}^{0}}{N}\frac{n_{R}^{A}}{N}}.$$If we introduce the dimensionless rates $\tau_{\left[AR\right]}:=\frac{\alpha }{\nu }$, which can be interpreted as the average attack rate in units of handling time, and $\tau_{\left[ARA\right]}:=\frac{\chi }{\eta }$, standing for the ratio between the pace of triplet formation and triplet degradation, respectively, we found that only in the limit of $\tau_{\left[ARA\right]}\ll 1$ and $\tau_{\left[AR\right]}\ll 1$, Equation (Equation 92) is a suitable approximation of Equation (Equation 91). The equation obtained is a predator-dependent functional response of the Beddington–DeAngelis type except for the product $\frac{n_{R}^{0}}{N}\frac{n_{R}^{A}}{N}$ in the denominator. Therefore, we note that only if the system is at a very high resource density ($n_{R}^{0}\approx N$), we would tend to observe the exact Beddington–DeAngelis functional form.

## 6. Discussion

The aim of statistical physics is to develop phenomenological macroscopic results from a probabilistic examination of the underlying microscopic processes. This allows connecting processes at different levels of coarse-grained description. We applied this approach to consumer-resource dynamics, connecting elemental processes of growth and consumption at the individual level to a more coarse-grained description that leads, in the limit of large populations, to a deterministic description in terms of the two-dimensional typical predator-prey model (the macroscopic law, sensu Van Kampen [35]).

We set up individual dynamics in a way that makes contact with classic models in ecology. For instance, if we consider only resource dynamics, we have a population that invades an area characterized by *N* sites with a given immigration rate per site, $\lambda_{R}$. Then, local growth occurs, that is individuals send a number of propagules per unit time that only settle down successfully in proportion to the number of free, still non-colonized sites. This means that the system can only pack a maximum number of individuals, *N*. Finally, individuals die at a density-independent per capita rate, $\delta_{R}$. Under the usual assumption of one individual per site, this model corresponds to the open Levins model [51,52,53,54]. The extension of this model to *S* resources that compete for the *N* sites has been also carefully analyzed in the literature [55,56], both using one-step stochastic process and in the deterministic limit. Furthermore, if we drop local growth, then we have a system of *S* species undergoing an independent immigration-birth-death process, which, under ecological equivalence, leads to a typical neutral model for biodiversity [57,58]. Interestingly, when we sample a total of $N_{S}$ individuals from such a neutral community dynamics, which are necessarily distributed in a vector of abundances $\vec{n}=\left(n_{1},\ldots ,n_{S}\right)$, the probability of obtaining a given configuration vector, $\vec{n}$, follows exactly the Etienne likelihood function [59,60], the corner-stone of neutral biodiversity theory.

Consumer-resource dynamics was also set up in a general way although it clearly allows for further generalizations. First, we considered also that consumers immigrate from outside the *N*-site system at a per site rate, $\lambda_{A}$. While the system can only pack *N* resource individuals, in principle, there is no limit for the number of consumers. This number will be dynamically controlled by the resource level. The size of the system only directly limits resources. This differs from most approaches (e.g., [37,46,61,62]), but as we showed, it is not a problem to expand the system in terms of the size parameter *N* and recover, in this way, the corresponding deterministic rate equations in the large *N* limit. Some of the results we covered here, such as the existence of a Hopf bifurcations, can only be recovered when there is no external immigration of consumers. Since natural systems are most of the time open and subject to external inputs, this means that nice and stable consumer-resource dynamics (limit cycles) should be very difficult to observe in nature due to the stabilizing role of external immigration [63]. This does not preclude recovering stable cycles under experimental, close-to-immigration, controlled conditions [64,65].

The way we set up the subset of reactions controlling feeding dynamics has also the potential to clarify much of the discussion about predator-dependent functional responses of the Beddington–DeAngelis type [20,66,67,68]. We defined interference between consumers through structures that require multiple individuals to interact and stay together for an amount of time. Under this assumption, incidentally, we showed that, by considering only interference between handing consumers and free consumers, the exact Beddington–DeAngelis functional form cannot be fully recovered. We believe this point requires careful analysis that we will develop in a further publication.

The minimal set of reactions that we conceived reproduces consumer-resource dynamics in terms of four ODEs, Equations (20)–(23), in the deterministic limit. This involves the description of the different processes driving dynamics at the individual level. Our main result in this work has to do with the comparison between two alternative ways of deriving functional responses. First, an asymptotic approximation for the mean field version of the complete set of reactions, borrowing slaving arguments from [46], gives rise to a consumer-resource model, which predicts a functional form for the average per capita rate at which consumers deplete resources. Secondly, in a rather classic way [14,19], we derived the functional responses by describing the setting of a typical feeding experiment, that is the same system in chemostatic conditions, by only considering the subset of reactions characterizing the feeding interaction. In the first case, when going from the four-equation to the two-equation system, the dynamics of the usual kind, i.e., predator-prey equations with Holling type I (Equations (47) and (48)), type II (Equations (50) and (51)), and Beddington–DeAngelis (Equations (55) and (56)), are recovered by carefully assuming separation of time scales in the processes involved. Moreover, depending on the particular assumptions behind the separation of time scales, different functional forms may arise. In general, our analysis showed that predator feeding rates in the 2D consumer-resource models do not exactly match the typical functional responses presented in the literature, which we mostly recovered here under chemostatic conditions. Therefore, our results clearly demonstrated a discrepancy between the functional responses that emerge from the analysis of the full consumer-resource system through asymptotic derivations and separation of time scales and the functional responses derived under chemostatic conditions. For example, the attack rate and the handling time of type II functional response obtained under chemostatic conditions differ from the same parameters of the type II functional response that results from asymptotic derivations.

Although finding effective low-dimensional representations from high-dimensional dynamical systems is not an easy task, which formally requires methods from singular perturbation theory [69,70], we believe that the origins of the discrepancy we report is not related to the separation of time scales we assumed in order to go from the full, four-equation ODE system to the typical predator-prey 2D model. We rather think that it has to do with the simple way in which consumers transform resources into new consumers, the so-called numerical response of consumers in the ecological literature [48]. The instantaneous coupling between consumption and reproduction we assumed here, which is also assumed in simple predator-prey models, is not realistic and might be at the basis of this inconsistency. Structured population models, which assume that consumers accumulate mass or energy as they feed and reproduce at a later stage [71], may shed new light on the relation between individual feeding dynamics and macroscopic functional responses. Other possible generalizations, such as the explicit consideration of spatial dynamics [67] (see also Appendix C), may also help explain the relation between individual feeding dynamics and coarse-grained descriptions leading to effective functional responses. Future avenues of research include developing more robust methods to make inference from experiments of consumer-resource systems based on stochastic dynamics [26,72,73] rather than on simple ODE systems.

In sum, we demonstrated that the use of simple predator-prey models to analyze both experiments and natural predator-prey interactions should be done with caution. This includes food-web theory and applications. Models are simple representations of reality. As such, it is not rare to find inconsistencies when we try different types of models to analyze the same system. The art comes when theory is able to clearly establish the range of situations in which a certain mathematical model matches reality, which allows for both predictions and further experiments. However, a successful model of this kind should never claim that the processes considered occur in exactly the same way as they do in nature. Most of the time, process representation is a strong idealization. In this sense, we believe that ecological theories, and probably theories in general, are always effective theories.

## Figures and Tables

**Figure 1 entropy-23-00575-f001:**
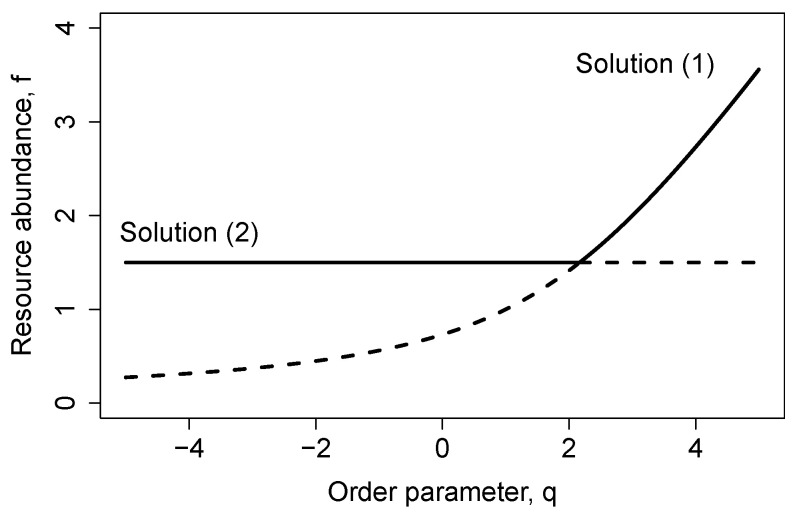
Transcritical bifurcation for $\lambda_{R}=2\beta$, $\lambda_{A}=0$, and $\delta_{A}=3\alpha /2$. Here, the vertical axis stands for the scaled resource abundance $f_{R}$, and the horizontal axis represents the control parameter $q=1-\delta_{R}/\beta$. For $q<q^{⋆}=13/6$, the (upper) solution $f_{R}^{\left(2\right)}=3/2$ is stable and for $q>q^{⋆}$ stability changes and $f_{R}^{\left(1\right)}=\frac{1}{2}\left(q-2+\sqrt{{\left(2-q\right)}^{2}+8}\right)$ becomes the stable resource scaled abundance. Observe that for $q>q^{⋆}$, consumer, pair, and triplet abundances become equal to zero in the stable branch. Stable (unstable) solutions are marked with full (dashed) lines.

**Figure 2 entropy-23-00575-f002:**
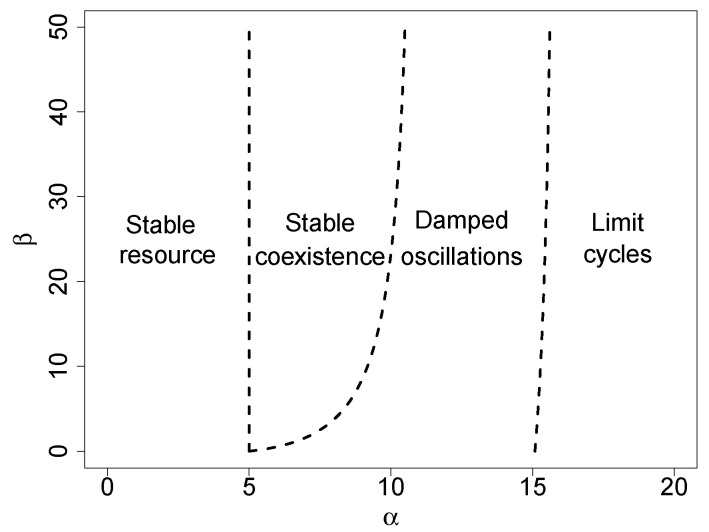
Parameter space showing transitions between different stability regimes for $\lambda_{R}=0$, $\lambda_{A}=0$, and $\delta_{R}=0$. In this case, the transcritical bifurcation threshold (Equation 32) reduces to the vertical line $\alpha =\delta_{A}$. The threshold $\alpha_{c}$ (cf. Equation (Equation 42)) for the Hopf bifurcation is the limit (for β→0) of the line that separates stable oscillations and limit cycles. The remaining lines were computed by numerical evaluation of the Jacobian eigenvalues.

**Figure 3 entropy-23-00575-f003:**
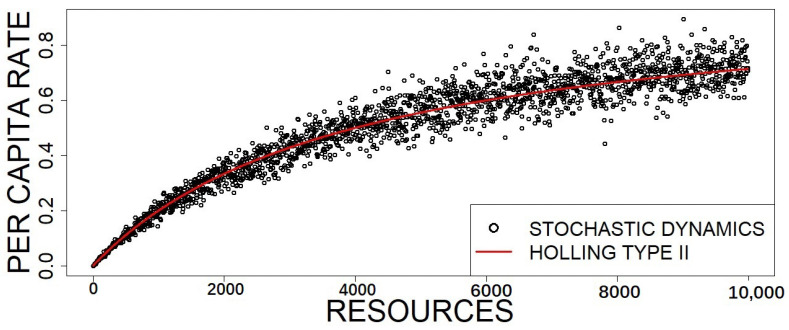
Plot of the per capita feeding rate of the stochastic process defined by the reactions (57)–(58). Simulations were carried out using the Gillespie algorithm with 10,000 steps, while the rates (black dots) were calculated, after a transient of 5000 steps, for different values of resource concentrations and compared with the functional response given by Equation (72) (red line). The simulation parameters are β=1.5, $\delta_{A}=1$, α=2.5, ν=1, and N= 10,000. The total number of consumers is fixed to $n_{A}^{0}=n_{A}+n_{\left[AR\right]}=200$.

**Figure 4 entropy-23-00575-f004:**
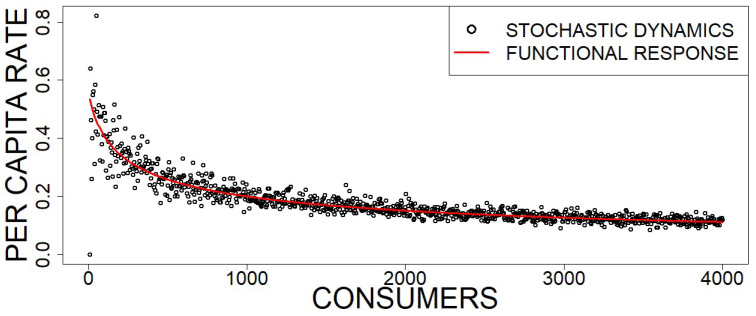
Plot of the per capita feeding rate of the stochastic process defined by the reactions (80)–(83). Simulations were carried out using the Gillespie algorithm with 15,000 steps, while the rates (black dots) were calculated, after a transient of 7500 steps, for different values of consumers concentrations, and compared with the functional response given by Equation (Equation 91) (red line). The simulation parameters are β=1.5, $\delta_{A}=1$, α=2.5, ν=1, η=1, χ=100, and N= 10,000. The total number of resources is fixed to $n_{R}^{0}=5000$.

## Data Availability

The code to reproduce simulated data and figures can be found at https://github.com/Gpalam/The_Stochastic_Nature_of_Functional_Responses.

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
