# Peer review of "The Stochastic Nature of Functional Responses"

_entropy, 2021, doi:10.3390/e23050575_

Round 1

Reviewer 1 Report

The Authors compare two alternative ways of deriving functional response. They perform an asymptotic approximation for the mean field approach of a set of individual interactions and compare with the classical derivation of the system under hemostatic conditions. A discrepancy between both derivations is found. The Authors claim that the discrepancy is due to the way in which consumers transform resources into new consumers. Form the physical point of view is an example of application  of the system size expansion by van Kampen to the problem of functional responses. The work is  well written and interesting both for the physical and ecological audience. I recommend publication in its present form.  

Author Response

Thank you reviewer 1 for your positive feedback.

Reviewer 2 Report

The report is attached in PDF format. 

Author Response

Please find attached our point to point reply.

Reviewer 3 Report

This is an interesting paper. The authors present two ways of deriving functional responses, and reveal that classical functional response parameters in effective 2D consumer-resource dynamics differ from the same parameters obtained by deriving functional responses for typical feeding experiments under chemostatic conditions. I have one suggestion. The authors should polish their paper, there are several typos.

Author Response

Thank you reviewer 3, we have polished the paper, corrected typos and made checked all equations.

Round 2

Reviewer 2 Report

The authors have considered all my suggestions. Now the presentation of the paper has improved. Therefore, I recommend the manuscript for publication. 

Three minor comments: 

  1. Please include "derivation of functional responses" in the title of section 4. This is what I mentioned in the last report, point 6. 
  2. In Figs. 3 and 4, the legend "stochastic dynamics" should be shown by the same symbol (a circle instead of a line) used in the plot. 
  3. I am happy that the authors have shown a simple derivation for type-III functional response. I suggest the authors to include this finding in the introduction.
